# Investigating the potential of aggregated mobility indices for inferring public transport ridership changes

**Maximiliano Lizana**[1,2], **Charisma Choudhury**[1]*, **David Watling**[1]

1 Institute for Transport Studies, University of Leeds, Leeds, United Kingdom, 2 Department of Civil Engineering, Universidad de La Frontera, Temuco, Chile

* c.f.choudhury@leeds.ac.uk

## Abstract

Aggregated mobility indices (AMIs) derived from information and communications technologies have recently emerged as a new data source for transport planners, with particular value during periods of major disturbances or when other sources of mobility data are scarce. Particularly, indices estimated on the aggregate user concentration in public transport (PT) hubs based on GPS of smartphones, or the number of PT navigation queries in smartphone applications have been used as proxies for the temporal changes in PT aggregate demand levels. Despite the popularity of these indices, it remains largely untested whether they can provide a reasonable characterisation of actual PT ridership changes. This study aims to address this research gap by investigating the reliability of using AMIs for inferring PT ridership changes by offering the first rigorous benchmarking between them and ridership data derived from smart card validations and tickets. For the comparison, we use monthly and daily ridership data from 12 cities worldwide and two AMIs shared globally by Google and Apple during periods of major change in 2020–22. We also explore the complementary role of AMIs on traditional ridership data. The comparative analysis revealed that the index based on human mobility (Google) exhibited a notable alignment with the trends reported by ridership data and performed better than the one based on PT queries (Apple). Our results differ from previous studies by showing that AMIs performed considerably better for similar periods. This finding highlights the huge relevance of dealing with methodological differences in datasets before comparing. Moreover, we demonstrated that AMIs can also complement data from smart card records when ticketing is missing or of doubtful quality. The outcomes of this study are particularly relevant for cities of developing countries, which usually have limited data to analyse their PT ridership, and AMIs may offer an attractive alternative.

## 1 Introduction

### 1.1 Public transport demand data

The availability of suitable data is critical for city planners to tackle the current and future challenges in urban mobility. This need is amplified when there is a disruptive change in urban

**Data Availability Statement:** The data sets used in this study are available in the Supportive Information files.

**Funding:** The funding for this research has been provided by the Chilean Agency of Research and Development (ANID) through the Becas Chile

scholarship (URL: https://www.anid.cl). Professor Charisma Choudhury's time was supported by the UKRI Future Leader Fellowship [MR/T020423/1] (URL: https://www.ukri.org/). The authors state that there was no additional internal/external funding received for this study. The funding and source of support had no role in study design, data collection and analysis, decision to publish, or preparation of the manuscript.

**Competing interests:** The authors declare that they have no known competing financial interests or personal relationships that could have appeared to influence the work reported in this paper.

mobility at any scale, ranging from local short-term events such as natural disasters, social unrest, and transport supply breakdown to global long-term events such as pandemics/epidemics, economic crises and conflicts. In this context, a continuous monitoring of public transport (PT) demand changes is essential for authorities and PT operators [1,2]. In spite of the growth in the availability of higher quality data in many parts of the world, still there remain many cities that do not have access to proper data for a constant characterisation of the PT demand; or even if they have it, the available data present limitations in terms of the quality and coverage. In cities without automated data collection systems to passively record ticketing levels, traditionally, the information related to PT demand has come from datasets that have been manually collected on a small population sample. Such data, despite providing granular information, has been criticised for the lack of feasibility to be steadily applied during long periods [3]. This makes them unsuitable to analyse dynamic PT demand changes and to quantify the impacts of unexpected disruptions [4–6]. By contrast, cities that have already adopted automated fare collection (AFC) schemes have had the advantage of analysing their PT demand information from smart cards and digital transactions [7–9]. However, some limitations on the fare collection system may affect the quality of these data [10]. For example, ridership data may be lower than the actual one when ticketing are missing or incomplete, such as in the cases of ticket-free riding days or when there are special periods where fare evasion is potentially higher. Additionally, AFC systems may only cover a limited number of the PT modes present in a city (e.g. metro rails only), capturing ridership data only of those modes [11,12]. In these cases, even cities with AFC systems can benefit from secondary data sources to complement traditional ones.

## 1.2 Aggregated mobility indices

The increasing penetration of Information and Communication Technologies (ICT) in society has allowed several emerging datasets to be harnessed to face urban mobility challenges [13]. Call detail records (CDRs) [6], social media data [14–16], Wi-Fi and Bluetooth traces [17], and web-based ticket records [18] are some of the technologies explored in the last decade to understand the behaviour of PT passengers. Despite the effort to leverage these data to study different characteristics of PT demand, their adoption has been mainly limited to research purposes and a few case studies, as such data availability remains largely restricted [3]. Less attention, however, has been paid to the usage of data sets associated with GPS traces collected by global mobile phone apps or the level of queries in travel planner apps in the PT sector [3,19]. This situation changed in 2020, following the urgent need of health authorities, local governments, transport agencies, and the public for continuously updated and easily accessible data to deal with the COVID-19 pandemic.

Aggregated mobility indices (AMIs) based on ICT were globally provided by tech companies during the COVID-19 pandemic to describe human mobility patterns in cities. AMIs were based on data collected from the regular use of mobile devices associated with GPS and apps, technologies that were already part of tech companies' products and services [20]. The information was aggregated to describe human mobility behaviour within cities, offering a near-complete coverage of the urban grid and a large proportion of the population. AMIs were used to analyse mobility trends and scenarios, and assess the effectiveness of mobility restrictions on human mobility [5,21–25]. AMIs were also employed in studying COVID-19 transmission [26], pandemic indicators [26,27], air quality [28,29] and economic recovery [30], among other topics. Big Tech companies such as Google and Apple shared reports on the aggregated mobility changes of the population at a city or regional scale between 2020 and 2022 [31,32]. Other companies, such as Moovit and Citymapper, which run travel planner apps, also offered similar mobility indices [33,34].

Among the AMIs proposed, Google COVID-19 Community Mobility Reports (GCMR) and Apple Mobility Trend Reports (AMTR) were the most popular. GCMR were based on the variation of human movements across different categories of locations (residential, workplace and public transport stations, among others) [35]. To measure the mobility changes related to PT, GCMR considered the access frequencies and the time spent on PT hubs (bus stops, train stations, etc.). The relative change was estimated by comparing a mobility level with a pre-pandemic baseline value. Some uses of the GCMR's PT index were the characterisation of the use of PT, the clustering of cities with similar PT demand change levels, and the assessment of the effectiveness of mobility restrictions [4,12,21,36–38]. On the other hand, AMTR reported indices estimated based on navigation data from the Apple Maps app service to describe its users' mobility trends [32]. AMTR showed daily relative changes for three transport modes (PT, walking and driving) by estimating the quotient between the volume of direction requests for a specific day and pre-pandemic baseline [20]. The characterisation of the change in mobility was one of the main uses of this data set [4,21,39–41].

## 1.3 Ridership data versus AMIs

Despite the widespread use of the AMIs provided by tech companies during the last three years, it is surprising that limited evidence of the reliability of these indices to represent actual PT demand shifts is available. As the importance of mobility data availability transcends the COVID-19 pandemic, a proper assessment of the potential of AMIs in PT is desirable for wider applications. So far, comparisons between AMIs that offered proxies for PT and ridership data have been provided tangentially by a few studies that analysed both data sources when characterising COVID-19's impact on PT demand. These studies preliminarily reported that AMI captured the generalised drop in ridership during the pandemic outbreak and that after it, they overestimated PT demand recovery [34,42]. For instance, using ridership data, a study conducted in Sweden [42] reported a reduction in PT demand of 40% in Skåne, 50% for Västra Götaland and 60% for Stockholm at the end of June 2020. By contrast, using the PT index of GCMR, the same study observed only a 0%, 10%, and 20% reduction in ridership, respectively. When they explored the PT index of AMTR, they obtained a reduction of around 20% with no noticeable difference between those areas. A smaller difference was observed in New York, where a 50% ridership decrease was observed using the PT index of AMTR when a 70% reduction was reported by the subway transactions [11]. In a study conducted on the Community of Madrid also for 2020, the authors contrasted smart card records with the Moovit mobility index. They found that during the recovery stage, the Moovit index reported a drop of only 5% compared to a reduction of 50% recorded for the ridership data [34]. Despite this evidence, several limitations in the existing studies lead to inconclusive findings about the level of accuracy of AMIs in terms of replicating PT ridership changes and their potential for wide-spread use in PT planning and operational decisions:

1. Early comparisons overlooked differences in the methodological approaches used to estimate AMIs. Therefore, the benchmarking required for properly comparing the datasets is yet to be conducted.

2. As the primary goal of the above-mentioned studies was to describe PT demand changes and not to assess the similarity between ridership data and AMIs, they did not conduct a formal quantitative comparison, limiting the current evidence to point-temporal comparisons and visual inspections of the trends only. In addition, as these early insights are based on data from the first half of 2020 and a few isolated contexts, there is a significant gap in the literature in studying a more comprehensive period and a wider sample of cases.

3. To the best of our knowledge, attempts to leverage the complementary role of AMIs on traditional ridership data have yet to be done (e.g., fill in temporal gaps in the data, identify supplementary information, etc.).

To address these gaps, this study aims to conduct a comprehensive similarity evaluation between the changes reported by AMIs for PT demand and ridership data. Monthly ridership data from 12 cities worldwide from eight countries and daily ridership for three case studies (London, New York and Santiago de Chile) were used for the analysis. Similarity metrics assessed the agreement between AMIs and ridership data for the period 2020–2022. Seasonal ARIMAX models were also employed to test the capacity of AMIs to predict PT demand changes in periods where ridership data did not record the actual demand. The results of this study provide a more comprehensive understanding of similarities and differences between the two data sources and reveal the potential role of AMIs in PT demand characterization, particularly in developing countries.

The remainder of this paper is structured as follows. The methodology of this study is provided in Section 2, including a description of the data and a definition of the metrics used to measure the degree of similarity between ridership data and AMIs. Section 3 shows the results of the similarity comparison and Section 4 presents the complementarity analysis between AMIs and ridership data. Finally, the implications of the findings and future perspectives are discussed in Section 5.

## 2 Methodology

This study investigates the reliability of using aggregated mobility indices (AMIs) for inferring PT ridership changes. Fig 1 shows the methodological procedure followed in this study. First, we retrieved data on AMIs and ridership data between 2020 and 2022 for several cities. Then, a common baseline was defined and adopted, allowing the comparison between data sets. AMIs and ridership were then analysed, and practical applications were explored. A detailed definition of each step is presented next.

### 2.1 Data

Two AMIs that offered proxies for PT were retrieved to be tested in their alignment with ridership changes. We selected Google COVID-19 Community Mobility Reports (GCMR) and Apple Mobility Trends (AMTR) as they offered global coverage and the most prolonged availability [31,32]. Additionally, they present proxies for PT use based on different ICT sources: GCMR used GPS traces from smartphones, and AMTR employed the queries for PT made in the Maps application of Apple devices (a further description of these indices is provided in Section 1.2). In this work, we will use the term Human Mobility Index (HMI) to refer to the

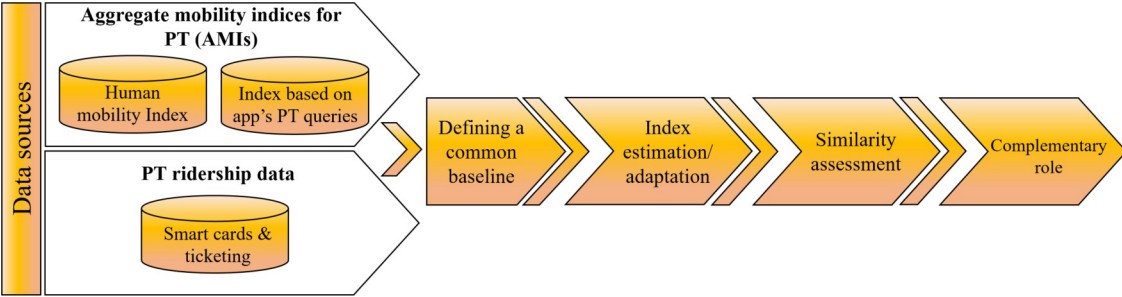

**Fig 1. Methodological approach followed in this study.**

**Table 1. Case studies for the comparison between AMIs and ridership data.**

| Country | City | Ridership PT authority | Google's Index (HMI) Spatial definition | Apple's Index (QI) Spatial definition |
|---------|------|------------------------|------------------------------------------|----------------------------------------|
| | | **Case studies with *daily* ridership data** | | |
| U.K. | London | Transport for London (TfL) | City of London / Sub-region 2 | London / City |
| U.S. | New York | MTA New York | New York County / Sub-region 2 | New York / City |
| Chile | Santiago | Met. Public Transport Agency (DTPM) | Santiago Province / Sub-region 2 | - |
| | | **Case studies with *monthly* ridership data** | | |
| Australia | Sydney | Transport for NSW | City of Sydney / Sub-region 2 | Sydney /City |
| Canada | Toronto | Toronto Transit Commission | Toronto / Sub-region 2 | Toronto / City |
| Colombia | Bogotá | Transmilenio* | Bogota / Sub-region 1 | - |
| U.S. | Dallas | Dallas Area Rapid Transit | Dallas County / Sub-region 2 | Dallas / City |
| U.S. | Denver | Regional Trip District | Denver County / Sub-region 2 | Denver / City |
| U.S. | Salt Lake | Utah Transit Authority | Salt Lake County / Sub-region 2 | Salt Lake / City |
| U.S. | Chicago | Chicago Transit Authority | Cook County / Sub-region 2 | Chicago / City |
| Taiwan | Taipei | Metro Taipei** | - | Taipei City / Sub-region |
| Hong Kong | Hong Kong | MTR Hong Kong** | Hong Kong / Country-region | - |

\* Only BRT ridership available.

\*\* Only metro ridership available.

particular index in the GCMR that measured the changes in human mobility in PT hubs (train stations, bus stops, etc.). Analogously, we will use the term Apple Query Index (QI) to refer to the category of AMTR that compared the level of queries for PT directions in Apple Maps. Both indices were updated daily from 2020 to 2022. Specifically, HMI was provided from 15 February 2020 to 15 October 2022 and QI from 13 January 2020 to 12 April 2022 (AMTR data for 11–12 May 2020, 12 March 2021 and 21 March 2022 were unavailable). On the other hand, ridership data came from validations made by smart cards and paper or digital tickets and were directly retrieved from the official portals of several PT operators. The inclusion criteria for selecting an urban area as a case study considered the availability of ridership data and AMIs. A total of 12 different case studies were selected (considering a monthly temporal resolution of ridership), aiming to include different contexts and increase the generalization of the similarity assessment. In only three of them, daily ridership data were publicly available (London, New York and Santiago de Chile). Both, *monthly* and *daily* ridership data were used in the similarity analysis (See S1 Table). Table 1 specifies the case studies included in the analysis, indicating the availability of AMIs as well as the temporal and spatial resolution of the retrieved ridership data. This work employed publicly available data, whose use complied with the terms and conditions for each source. Further details of the terms and conditions can be found directly in the web pages of each source using the links provided in the Supporting files (S1 Table).

## 2.2 Establishing a common definition

To generate comparable datasets, a common basis for estimating mobility changes was adopted. This common basis was required due to differences in how AMIs were reported and the absolute nature of the ridership data (i.e. the total number of transactions), aspects that were ignored in early comparisons. The GCMR and AMTR reported daily relative changes by estimating the quotient between the mobility volume for a specific day and a baseline mobility volume defined for the pre-pandemic. The proportion obtained was reported as a percentage, and a positive value indicated the percentage of increase with respect to the baseline, while a

**Table 2. Details of the common baseline adopted to estimate relative changes in ridership data and the QI.**

| | | Apple's Index[1] (QI) | | Ridership Data | | |
|---|---|---|---|---|---|---|
| | | London | New York | London | New York | Santiago |
| Period data available | | 13 Jan 2020 to 12 Apr 2022 | | 01 Jan 2020 to 31 Oct 2022 | 01 Mar 2020 to 31 Oct 2022[2] | 01 Jan 2020 to 31 Oct 2022 |
| **Baseline definition (pre-pandemic)** | | | | | | |
| Original reported values | | AMTR | | Recorded subway and bus ridership, nominal values | Recorded subway and bus ridership, nominal values | Recorded subway and bus ridership, nominal values. |
| Common baseline definition applied | | The median value for each day of the week over the three weeks between 13 January and 2 February 2020 | | The median value for each day of the week over the five weeks between 3 January and 6 February 2020 | Average value for January 2020 for weekdays, Saturday and Sunday | The median value for each day of the week over the five weeks between 3 January and 6 February 2020 |
| **Consistency of baseline values** | | | | | | |
| Coeff. Var. baseline values | Mon | 1.7% | 5.5% | 3.9% | - | 1.9% |
| | Tue | 1.8% | 1.4% | 3.3% | - | 2.3% |
| | Wed | 2.2% | 2.3% | 2.7% | - | 4.6% |
| | Thu | 2.4% | 2.4% | 2.3% | - | 4.3% |
| | Fri | 3.2% | 2.6% | 1.0% | - | 4.8% |
| | Sat | 1.4% | 3.2% | 6.2% | - | 5.0% |
| | Sun | 2.9% | 4.9% | 5.0% | - | 5.0% |

[1] The category of PT of the AMTR was not available for Santiago de Chile.

[2] No bus ridership data collected directly from a smart card or ticketing validation was available between 1 Mar and 30 Sep 2020.

negative one specified the degree of reduction. However, the baseline definition adopted by each index was different. As the QI employed as a baseline the number of queries of only one day, this index was more susceptible to high variability due to the lack of inclusion of weekly mobility cycles. In the case of ridership data, a normalisation of their values respecting a baseline was also required to obtain a relative scale. A review of the criteria employed in the literature for estimating relative changes with aggregate PT demand supported the baseline definition adopted by the HMI [34,43], and for this reason, it was employed as the consistent basis in the current paper. The definition includes the choice of a pre-COVID-19 period, the recognition of demand variability within the week and a way to deal with potential outliers. As no AMIs were available before 2020, using data from 2019 to describe the pre-pandemic period was not possible. Both ridership data and the QI were adapted according to the HMI's baseline definition. We present the details of the baseline definitions adopted for the case studies where daily ridership was available in Table 2. Table 2 also explores the consistency of daily values for ridership and the QI in the baseline period. Coefficients of variation smaller than 6.0% were observed, revealing high stability in the mobility trends of the same days of the week for the period that characterised the pre-pandemic. It was interesting to observe also for this period that the QI depicted the highest demand for PT information on Fridays and Saturdays, contrasting with the typical daily variability of ridership data (See S1 Fig).

The expressions used to apply the common basis on ridership data and the QI are presented in Eqs 1 and 2. We call these new indices relative ridership change index (RRC) and Apple

query modified Index (QMI). The RRC at a time $t$ ($RRC_t$) was defined as follows:

$$RRC_t = \left( \frac{r_t}{R_{f(t)}} - 1 \right) \cdot 100 \tag{1}$$

Where $r_t$ is the ridership on day $t$ ($t = 1, 2, \ldots, T$) and $R_{f(t)}$ is the baseline ridership for each day of week $f$, whose value in Eq 1 depends on the day of the week corresponding to $t$. If both $r_t$ and its corresponding $R_{f(t)}$ were the same, the quotient is one and the $RRC_t$ is equal to zero (i.e. 0% change). The RRC takes a negative value if the ridership in the time $t$ is smaller than the one existing in the baseline period for the corresponding day of the week. For instance, if the ridership were half compared with the baseline value, the index would be equal to -50 (%). The same interpretation apply for the HMI. In the case of New York, daily ridership was not available for January. Therefore, we use the reported average value of ridership for weekdays, Saturdays and Sundays during January 2020 as baseline values. To estimate RRC for the case studies where only monthly ridership was available, we first estimated the average daily ridership for each month, dividing monthly ridership by the number of days of each month. Then, the RRC was estimated analogously, employing the average daily ridership of January 2020 as a baseline value.

In the case of the QI, its original values were reported as percentage changes relative to one particular day. Apple provided these values on a base of 100, assigning a value of zero to the QI of 13 January 2020. Then a value of 5.0 would indicate that for a particular day, the number of queries was five percent higher than the base day. The adoption of the common baseline for estimating relative changes for this index was addressed by proposing the Apple's query modified index (QMI):

$$QMI_t = \left( \frac{QI_t + 100}{QIB_{f(t)} + 100} - 1 \right) \cdot 100 \tag{2}$$

Where $QI_t$ is the Apple's query index on day $t$ ($t = 1, 2, \ldots, T$) and $QIB_{f(t)}$ is the Apple's query index baseline for each day of the week $f$, whose value in Eq 2 depends on the day of the week corresponding to $t$. Each $QIB$ was estimated as the median $QI$ value for the same day over the three weeks between 13 January and 2 February 2020 (as there were no earlier values). In this way, the QMI overcomes the original limitation of the QI, comparing query levels of the same days of the week and controlling for outliers if they were present. The QMI shares the same interpretation with HMI and RRC. Monthly QMI and HMI were obtained by averaging their daily values for each month.

## 2.3 Similarity assessment

The degree of similarity between the values reported by AMIs (HMI and QMI) and RRC was assessed by applying metrics under a time series approach. For the monthly analysis, we included the mean Euclidean distance (MED), the cosine distance (COS) and a trend similarity index (TSI). The Dynamic Time Warping distance (DTW) and the Granger Causality test were included for the daily similarity analysis where a higher granularity in the data was available. The MED is recommended when a straightforward interpretation of the differences is required. In our case, as the time series values are all relative changes (%), the MED interpretation is the average distance in percentage points between the relative change reported by the AMIs (HMI or QMI) and RRC. The COS is a similarity measurement between two vectors defined in an inner product space [44]. Their values are always between -1 and 1, where 1 means perfect alignment and -1 indicates the opposite. DTW is an alignment-based metric

that estimates the Euclidean distance between two time series that may not be aligned [44]. We included DTW for the daily analysis to deal with potential shifts in the times series, particularly present in the QMI. The Granger Causality test determined whether AMIs could be used to forecast the RRC values [45]. This statistical hypothesis test uses Student's statistic and F-statistic tests to determine whether values of a certain variable provide statistically significant information about the values of Y. The trend similarity index (TSI) was estimated as the proportion of slopes with the same sign for the same pair of consecutive months/days between the RRC and AMIs. The sign of the slopes for two consecutive times was obtained by observing the direction of the change between the values of each index. The TSI between RRC and an AMI (HMI or QMI) was defined as:

$$TSI_{RRC,AMI} = \frac{\sum_t s_t^{RRC,AMI}}{T-1} \tag{3}$$

$$s_t^{RRC,AMI} = \begin{cases} 1, \Delta RRC_{t,t+1} \cdot \Delta AMI_{t,t+1} \geq 0 \\ 0, otherwise \end{cases} \tag{4}$$

Where $\Delta$ indicate the difference between two consecutives values for the respective index, and T is the length of the time series. TSI metric ranges between 0 and 1, where the value one means that the AMI replicated exactly the same direction of change of the RRC and zero the case of a complete disagreement.

## 2.4 Complementary role of AMIs

We also explore the complementary role of AMIs in contexts where ridership data did not capture the actual PT demand and on atypical days where mobility demand was extraordinarily high. Hence, the next situations helped illustrate the role that AMIs may play in complementing ridership data:

1. Free bus travel period in London during the pandemic outbreak: From 20 April to 30 May 2020, Transport for London introduced middle/rear-door-only boarding in bus services to take care of drivers. AMIs were used here to reveal an approximation of the actual PT demand in this period where ridership was under-reported.

2. Partial ridership data in New York MTA: No bus ridership data collected directly from a smart card or ticketing validation was available for the New York MTA between 1 March and 30 September 2020. Using AMIs, an approximation of the actual RRC in this period was estimated.

3. High mobility demand day for Santiago: The day of the national referendum in Chile (Sunday, 4 September 2022) was marked by an extraordinarily high mobility, resulting in the highest recorded RRC for Santiago in the study period. We assessed the discrepancies between the predicted RRC (based on the AMIs) and the recorded RRC.

Autoregressive Integrated Moving Average (ARIMA) models were employed to calibrate the relationship between the recorded RRC and the AMIs. For this, we selected the AMI that exhibited the highest similarity with RRC, while the calibration was made on periods with the most stable conditions available. ARIMA models are particularly efficient and appropriate when successive observations show serial dependence (e.g. in this case, daily observations), and therefore, the assumption of independent errors typically made for cross-section regression data is violated. At the same time, this modelling approach allows testing whether the AMI contribution to explain RRC is statistically significant. As a weekly periodicity was also

found in the descriptive analysis (see S2 Fig), an appropriate model specification may consider both, daily and weekly autocorrelation. To consider both correlations, a multiplicative seasonal ARIMA model is specified, where one component ($p$, $d$, $q$) captures the daily correlation, and a second component ($P$, $D$, $Q$) explains the weekly correlation in the data. If $s$ is the seasonal period of the time series (considering weekly seasonality $s = 7$), then the seasonal ARIMAX ($p$, $d$, $q$)×($P$, $D$, $Q$)[$s$] [46] can be written as follow:

$$\Phi_P^*(L^s)\Phi_p(L)(1 - L^s)^D(1 - L)^d y_t = \mu + \Theta_Q^*(L^s)\Theta_q(L)\varepsilon_t + \omega AMI_t \tag{5}$$

where $y_t$ is the value of the *RRC* time series for the time $t$. $\varepsilon_t$ is the white noise process (i.e. random error, i.i.d. Gaussian ($0$, $\sigma_\varepsilon^2$)) and $L$ is the backshift or Lag operator, defined as $Ly_t = y_{t-1}$. $d$ represents the differences that can be applied on the dependent variable to obtain a stationary time series for the non-seasonal model. $\Phi_p(L)$ is the polynomial of order $p$ that contains the marginal contribution of the auto-regressive (AR) component and $\Theta_q(L)$ the polynomial of order $q$ of the moving average (MA). $\Phi_P^*(L^s)$ is the operator of the seasonal AR component with order $P$, $D$ is the seasonal differences number and $\Theta_Q^*(L^s)$ is the operator for the seasonal MA with order $Q$. Note that we have added in the last term of Eq 5 the AMI, which is an exogenous variable in the modelling with coefficient $\omega$.

## 3 Exploratory analysis

### 3.1 Monthly analysis

A monthly analysis of 12 cities worldwide from eight countries showed that HMI and QMI were capable of replicating the RRC with different degrees of accuracy. Fig 2 presents the monthly variability of each index for the entire study period per case study, while Table 3 presents the results for the similarity metrics. Overall, AMIs correctly mimicked the main direction of changes depicted by RRC. In all the cases considered, AMIs properly replicated the drop in PT demand during the pandemic outbreak. However, in most cases, AMIs reported higher PT demand recoveries than the RRC. The average MED for the HMI and QMI were 11.9 and 11.6 for 2020 and 14.4 and 26.6 percentage points for the entire period, respectively. Cities like London and Sydney exhibited the greatest match between HMI and RRC, with small MED obtained (5.1 and 4.0). However, common MED were between 10 and 20 percentage points for most studied cases. Regarding the QMI, this index exhibited a similar adjustment to HMI until April 2021. After this date, QMI showed a general increase until August 2021, when the index stabilised around 60 percentage points above RRC values. The Similarity Trend Index (STI) ranged from 0.71 to 0.94 for the HMI and 0.73 to 0.92 for the QMI, revealing a high capability to replicate the direction of change of the monthly ridership trends by the AMIs. The cosine distance values supported these results showing magnitudes that indicate high similarity. Case studies where only partial ridership information was retrieved showed higher differences compared with the general trends. For instance, the greatest difference between HMI and RRC was observed in Bogota. This may be explained as ridership data for Bogota only describes the BRT system's demand and does not consider the local bus system. Moreover, contrary to the remaining case studies, AMIs in Taipei and Hong Kong reported lower PT demand recovery than the RRC. This difference may be explained by considering that only metro ridership was available for these two cities.

### 3.2 Daily analysis

The daily analysis compared HMI and QMI with RRC for the cities of London, New York and Santiago. Fig 3 provides a graphical illustration of the aggregate PT demand shifts depicted by

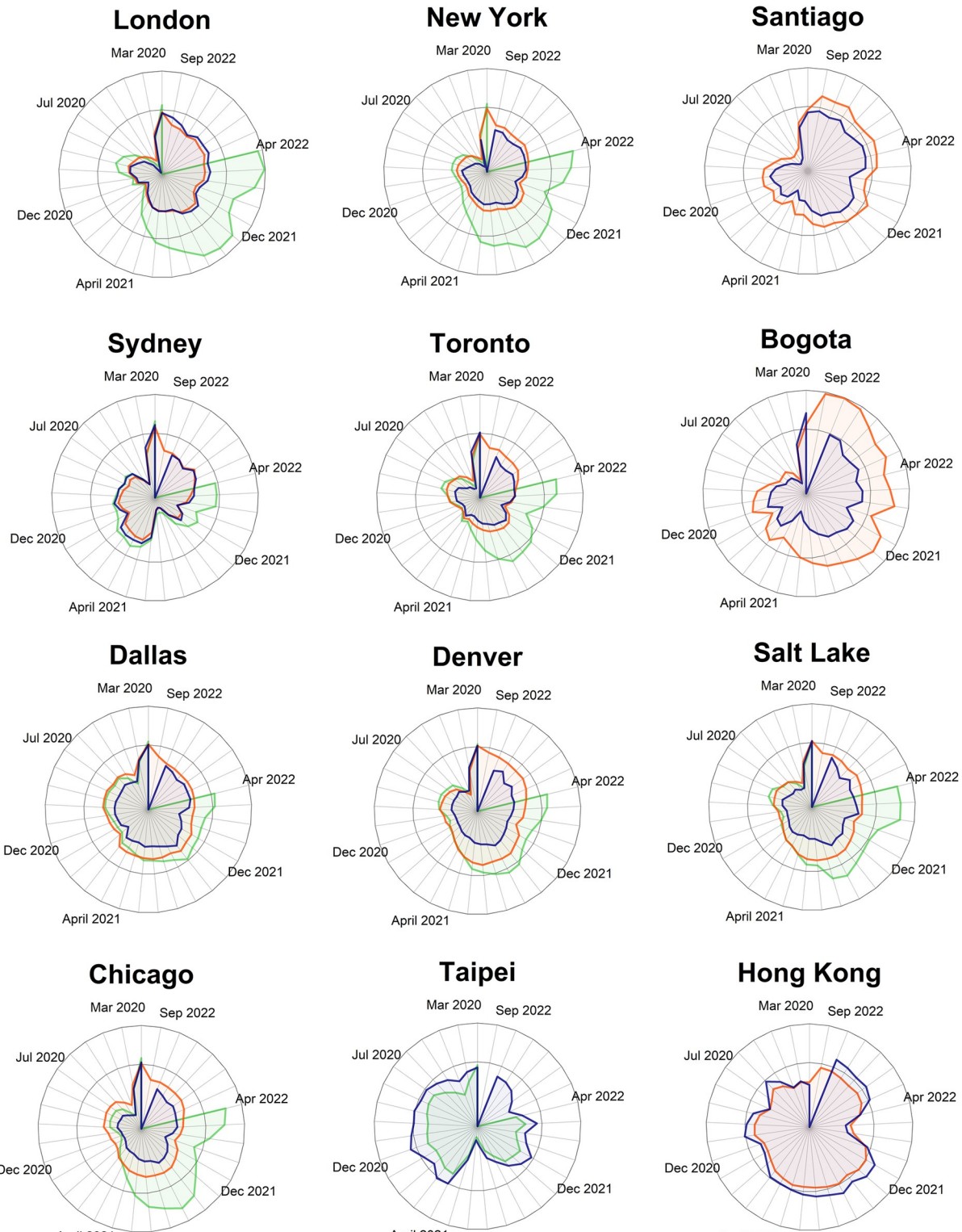

**Fig 2. Average monthly changes in HMI (orange), QMI (green), and RRC (blue).** Centre of the graphic indicates -100% change, central grid circumference 0% change and external grid circumference +60% change (all compared with baseline values). Data from February 2020 to October 2022 (for some case studies, available ridership data end in July 2022).

**Table 3. Monthly similarity metrics between AMIs and RRC.**

| Google's human mobility index (HMI) | | | | | | |
|---|---|---|---|---|---|---|
| | **MED** | | | | **STI** | **COS** |
| **Location** | All years | 2020 | 2021 | 2022 | All years | All years |
| London | 5.8 ●●● | 8.3 ●●● | 3.0 ●●● | 6.5 ●●● | 0.87 ▲▲△ | 0.99 ▲▲▲ |
| New York | 9.2 ●●● | 12.5 ●●○ | 8.2 ●●● | 7.0 ●●● | 0.93 ▲▲▲ | 0.99 ▲▲▲ |
| Santiago | 18.1 ●●○ | 14.2 ●●○ | 18.6 ●●○ | 21.8 ●●○ | 0.90 ▲▲▲ | 0.94 ▲▲▲ |
| Sydney | 4.1 ●●● | 5.3 ●●● | 4.3 ●●● | 2.1 ●●● | 0.94 ▲▲▲ | 0.99 ▲▲▲ |
| Toronto | 10.1 ●●○ | 13.9 ●●○ | 8.1 ●●● | 8.1 ●●● | 0.90 ▲▲▲ | 0.99 ▲▲▲ |
| Bogota | 35.2 ●○○ | 14.6 ●●○ | 43.0 ●○○ | 49.8 ●○○ | 0.84 ▲▲△ | 0.52 ▲△△ |
| Dallas | 13.4 ●●○ | 13.2 ●●○ | 16.7 ●●○ | 9.4 ●●● | 0.71 ▲△△ | 0.99 ▲▲▲ |
| Denver | 18.1 ●●○ | 10.3 ●●○ | 23.9 ●●○ | 20.0 ●●○ | 0.77 ▲△△ | 0.96 ▲▲▲ |
| Salt Lake | 15.7 ●●○ | 12.4 ●●○ | 21.4 ●●○ | 12.2 ●●○ | 0.71 ▲△△ | 0.97 ▲▲▲ |
| Chicago | 18.0 ●●○ | 18.8 ●●○ | 20.3 ●●○ | 13.8 ●●○ | 0.90 ▲▲▲ | 0.99 ▲▲▲ |
| Taipei | - | - | - | - | - | - |
| Hong Kong | 10.3 ●●○ | 7.4 ●●● | 11.9 ●●○ | 11.8 ●●○ | 0.94 ▲▲▲ | 0.81 ▲▲▲ |
| Average | 14.4 | 11.9 | 16.3 | 14.8 | 0.86 | 0.92 |
| Std. Deviation | 8.1 | 3.6 | 10.8 | 12.4 | 0.08 | 0.14 |
| Median | 13.4 | 12.5 | 16.7 | 11.9 | 0.90 | 0.99 |
| Apple query modified index (QMI) | | | | | | |
| | **MED** | | | | **STI** | **COS** |
| **Locations** | All years | 2020 | 2021 | 2022 | All years | All years |
| London | 36.2 ●○○ | 13.4 ●●○ | 44.6 ●○○ | 73.3 ●○○ | 0.81 ▲▲△ | 0.61 ▲△△ |
| New York | 37.6 ●○○ | 14.8 ●●○ | 47.7 ●○○ | 64.3 ●○○ | 0.88 ▲▲△ | 0.67 ▲△△ |
| Santiago | - | - | - | - | - | - |
| Sydney | 11.2 ●●○ | 2.8 ●●● | 10.9 ●●○ | 35.3 ●○○ | 0.81 ▲▲△ | 0.96 ▲▲▲ |
| Toronto | 28.4 ●○○ | 13.1 ●●○ | 33.3 ●○○ | 56.0 ●○○ | 0.88 ▲▲△ | 0.83 ▲▲△ |
| Bogota | - | - | - | - | - | - |
| Dallas | 18.6 ●●○ | 10.1 ●●○ | 19.9 ●●○ | 38.0 ●○○ | 0.77 ▲△△ | 0.90 ▲▲▲ |
| Denver | 27.0 ●○○ | 12.1 ●●○ | 33.5 ●○○ | 48.1 ●○○ | 0.92 ▲▲▲ | 0.83 ▲▲△ |
| Salt Lake | 30.1 ●○○ | 12.0 ●●○ | 35.3 ●○○ | 64.6 ●○○ | 0.73 ▲△△ | 0.67 ▲△△ |
| Chicago | 33.5 ●○○ | 7.6 ●●● | 47.1 ●○○ | 64.2 ●○○ | 0.85 ▲▲△ | 0.66 ▲△△ |
| Taipei | 16.8 ●●○ | 18.7 ●●○ | 13.5 ●●○ | 14.4 ●●○ | 0.85 ▲▲△ | 0.91 ▲▲▲ |
| Hong Kong | - | - | - | - | - | - |
| Average | 26.6 | 11.6 | 31.8 | 50.9 | 0.83 | 0.78 |
| Std. Deviation | 8.7 | 4.2 | 13.3 | 17.7 | 0.06 | 0.12 |
| Median | 28.4 | 12.1 | 33.5 | 56.0 | 0.85 | 0.83 |

AMIs: aggregate mobility indices, RRC: ridership relative change index, MED: mean Euclidean distance, STI: similarity trend index, COS: the cosine distance. ●●●: MED ≤ 10, ●●○: 10<MED ≤25, ●○○: MED>25. ▲▲▲: STI/COS ≥ 0.90, ▲▲△: 0.90>STI/COS≥0.80, ▲△△: STI/COS <0.80.

each index. The HMI was found to match surprisingly well with the daily RRC time series. At the same time, the QMI displayed a reasonable fit in terms of the magnitude of the PT demand recovery until the first half of 2021. For that period, the main trends of peaks and troughs of the RRC time series were also depicted appropriately by HMI and QMI, including short sharp reductions during holidays. A particular concurrence of the values of all indices was observed during the periods where the stricter mobility restrictions were in place (pandemic outbreak and from November 2020 to February 2021).

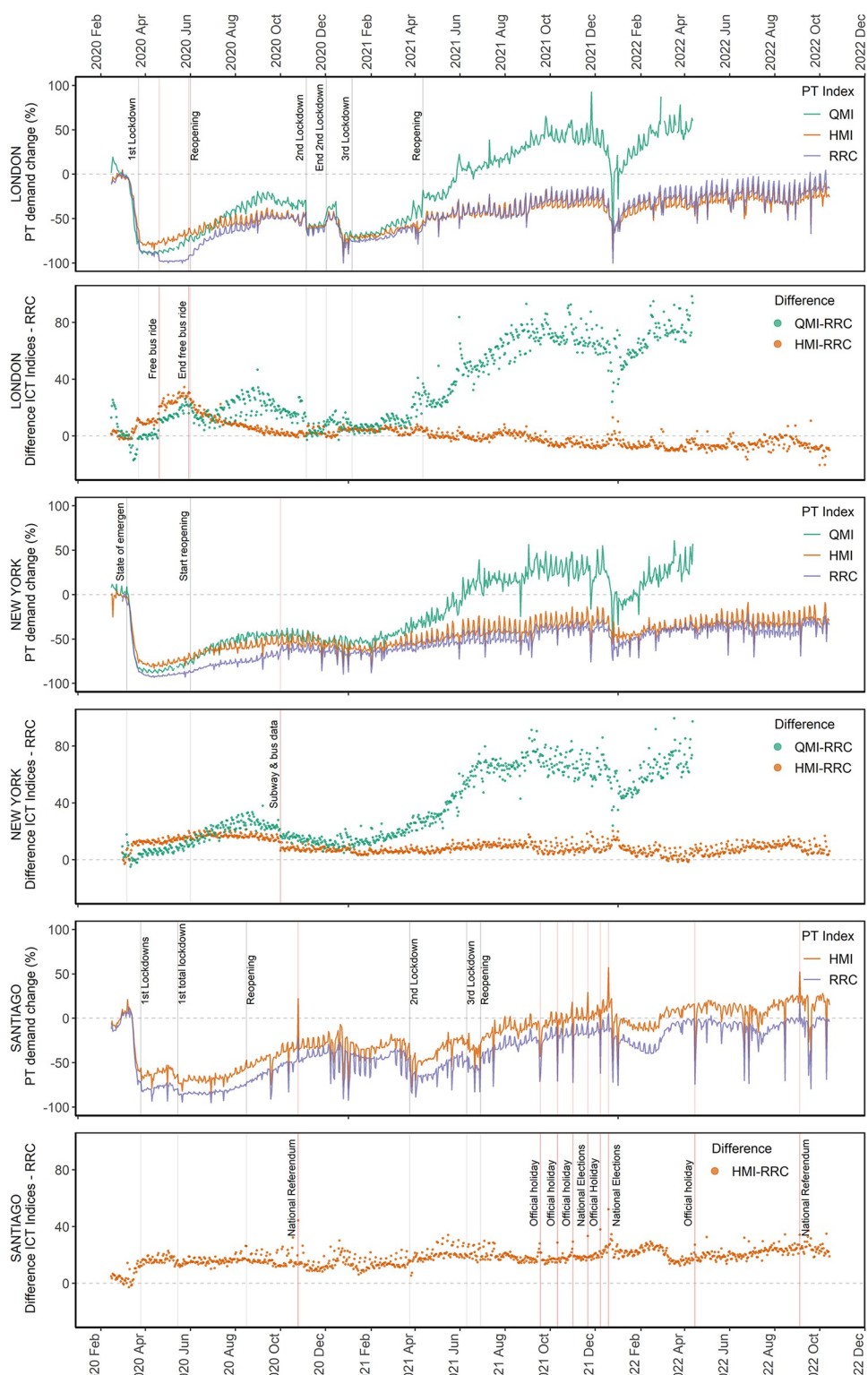

**Fig 3. Daily variation of PT mobility indices.** HMI, QMI and RRC for the case studies of London, New York and Santiago.

To identify changes in the pattern of the dissimilarity between the AMIs and RRC, Fig 3 also presents the differences between them for each day. A positive distance indicates that the AMIs observed a lower relative drop or a higher relative increase than the RRC. The difference between HMI and the RRC showed relatively constant values, whilst the difference between QMI and the RRC exhibited greater variability. In general, mostly positive differences were observed, except for London, where the HMI generated negative differences from May 2021 onwards. This situation coincides with changes in the fare scheme for children, which involved removing free travel for some ages. The highest dissimilarities in the London case between HMI and RRC were observed from April to May 2020, when mandatory payments in London's buses were suspended. In the case of New York, a change in the trend of the differences was observed during the first half of 2020, where RRC was represented only by the subway ridership. In the case of Santiago, the most remarkable observed differences were associated with sharp peaks in the HMI during special events (e.g. national elections and referenda). For the QMI, the highest differences with the RRC were observed during the first recovery period (June to November 2020) and during the second recovery (April 2021 onwards, where QMI was considerably higher). A ratio of increase in the number of PT queries greater than the recovery in the actual ridership during the first half of 2021, when many restrictions were eased, would explain these discrepancies. This interpretation is supported by the results of Sydney (see Fig 2), where its QMI experienced a similar increase when a similar ease of mobility restrictions started in that city at the end of 2021.

Similarity metrics presented in Table 4 support the descriptive analysis based on Fig 3. Considering the entire period, the MED for the HMI were 6.2, 9.1 and 18.2 for London, New York and Santiago, respectively. In the same order, standard deviation values of 5.8, 4.5 and 5.7 were calculated, indicating consistency among the case studies in terms of the differences between the HMI and the RRC. For each case study, the MED of HMI was seen to be relatively constant across the three years, showing great stability in its capabilities of replicating the changes reported by ridership data. In the case of the QMI, an overall MED of 35 percentage points was estimated for London and New York. Contrasting with the HMI results, the QMI showed an increasing difference along the time series with the RRC. Moreover, the standard deviation of the MED of QMI ranged from 26 to 28 percentage points, considerably higher than the one estimated for HMI. The analysis based on the DTW distance offered a similar interpretation for the AMIs. The Similarity Trend Index (STI) showed that the HMI replicated correctly between 75% and 85% of the RRC trend change directions; in the case of the QMI, the STI dropped to values from 54% to 74%. This may be explained considering that RRC and HMI depicted higher PT demand recovery on weekends, while QMI reported greater values on Fridays and Saturdays (see S2 Fig). The cosine distance indicated that QMI only presented a high similarity with RRC during 2020 and the Granger Causality Test indicated that both HMI and QMI could be used to predict RRC.

## 4 Modelling results

Based on the results of the similarity analysis, the HMI was selected as the best candidate for exploring the capability of AMIs to complement ridership data. Seasonal ARIMAX models were used to calibrate the relationship between HMI and RRC and then to forecast RRC for three particular cases (which have already been described in detail in section 3.4): a free bus travel period in London, a partial ridership data period in New York (both during the first half of 2020), and the day with the highest recorded RRC in Santiago during late 2022. A seasonal component of seven days was considered as the correlograms of the time series identified weekly periodicity. The periods used to fit and validate the models were selected considering

**Table 4. Similarity metrics between AMIs and RRC, daily analysis.**

| Period | Similarity measurement | Google' human mobility Index (HMI) | | | Query modified index (QMI) | |
|---|---|---|---|---|---|---|
| | | London | New York | Santiago | London | New York |
| **Distance between AMIs and RRC (in percentage points)** | | | | | | |
| All | Mean Euclidian distance (MED) | 6.2 | 9.1 | 18.2 | 35.7 | 36.1 |
| 2020 | | 8.8 | 12.0 | 14.7 | 13.6 | 14.5 |
| 2021 | | 3.5 | 8.2 | 18.6 | 44.9 | 47.9 |
| 2022 | | 6.6 | 7.1 | 21.2 | 72.0 | 61.9 |
| **Dynamic Time Warping distance (DTW)** | | | | | | |
| All | DTW | 5.0 | 5.8 | 9.6 | 24.5 | 24.2 |
| 2020 | | 4.7 | 5.2 | 6.6 | 7.4 | 6.3 |
| 2021 | | 3.8 | 6.1 | 9.4 | 30.5 | 24.2 |
| 2022 | | 6.9 | 6.4 | 16.8 | 64.6 | 60.3 |
| All | MED Std. Dev. | 5.8 | 4.5 | 5.7 | 28.5 | 25.2 |
| | MED Weekdays | 5.8 | 8.3 | 17.5 | 37.3 | 36.5 |
| | MED Weekends | 7.1 | 11.2 | 20.1 | 31.6 | 35.2 |
| **Similarity Trend Index (STI)** | | | | | | |
| All | STI | 0.80 | 0.83 | 0.75 | 0.61 | 0.73 |
| 2020 | | 0.80 | 0.82 | 0.68 | 0.59 | 0.70 |
| 2021 | | 0.77 | 0.83 | 0.76 | 0.63 | 0.73 |
| 2022 | | 0.85 | 0.83 | 0.83 | 0.54 | 0.74 |
| **Cosine distance (COS)** | | | | | | |
| All | COS | 0.99 | 0.99 | 0.94 | 0.62 | 0.69 |
| 2020 | | 0.99 | 0.99 | 0.99 | 0.98 | 0.99 |
| 2021 | | 0.99 | 0.99 | 0.94 | 0.37 | 0.30 |
| 2022 | | 0.98 | 0.99 | 0.81 | -0.78 | -0.55 |
| **Granger Causality Test** | | | | | | |
| 2020 to 2022 | Test F | 4.8 | 11.5 | 19.4 | 44.8 | 45.6 |
| | $p$ value | 0.03 | <0.01 | <0.01 | <0.01 | <0.01 |

the nearest interval to the research periods with homogeneous differences between the HMI and RRC. Two models were fitted for London, one with data located before the research period and a second with data after it. For fitting and validation purposes, the selected data were split into two segments considering a proportion 5:1. Details of the fitting, validation and research periods, as well as the modelling results, are shown in Table 5.

The results highlighted the statistical significance of the HMI in the model estimation of RRC (t-statistic higher than 50.0), whose relationship with the actual RRC ($\omega$) was estimated between 0.97 and 1.15. Both, the non-seasonal (daily correlation) and the seasonal (weekly correlation) components were statistically significant in the modelling. In the non-seasonal component, coefficients $\phi$ of the Autoregressive model (AR) were significant in the first order ($p = 1$). This means that the RRC on a day before ($t$-1) only was relevant to explain the RRC value of the next observation ($t$). AR coefficients were all positive, ranging from 0.24 to 0.96. In the case of the Moving Average (MA) coefficients ($\theta$), those associated with the MA of orders 1 and 2 were statistically significant. This implies the prediction benefited from correcting the error term of the lagged RRC prediction $t$-1 and $t$-2. Analogous results were observed for the seasonal component but related to observations of consecutive weeks (e.g. $\phi_1^*$ indicates a statistically significant effect of the RRC value of the previous week to the prediction of RRC of the next, considering the same day). The residual of the fitted models satisfied white noise

**Table 5. Seasonal ARIMAX time series modelling results.**

| | London | | New York | Santiago |
|---|---|---|---|---|
| **Period details** | | | | |
| **Forecasting** | *backward* | *forward* | *backward* | *forward* |
| **Fitting period** | 01 Dec 2020 to 30 Sep 2021 | 15 Feb to 07 April 2020 | 01 Dec 2020 to 30 Sep 2021 | 10 Mar to 10 Aug 2022 |
| **Validation period** | 01 Oct 2020 to 30 Nov 2020 | 08–14 April 2020 | 01 Oct 2020 to 30 Nov 2020 | 11 Aug to 03 Sep—05 Sep to 11 Sep 2022 |
| **Research period** | 15 April to 30 Sep 2020 | 15 April to 30 Sep 2020 | 01 March to 30 Sep 2020 | 04 Sep 2022 |
| **Modelling results** | | | | |
| **Variable** | **Coef (Test-t)** | **Coef (Test-t)** | **Coef (Test-t)** | **Coef (Test-t)** |
| Exogenous variable–Google human mobility Index (HMI) | | | | |
| **HMI ($\omega$)** | 1.15 (54.25) | 1.12 (64.86) | 0.97 (57.26) | 1.14 (51.14) |
| Model specification | | | | |
| **($p, d, q$)($P, D, Q$)** | (1,0,1)(1,1,1) | (1,0,0)(1,0,0) | (1,0,2)(1,0,1) | (1,1,1)(2,0,0) |
| Non-Seasonal Component ($p, d, q$) | | | | |
| **Intercept ($\mu$)** | - | -0.52 (0.40) | -9.31 (2.29) | - |
| **AR1 ($\phi_1$)** | 0.84 (15.21) | 0.64 (5.80) | 0.96 (43.43) | 0.24 (2.28) |
| **MA1 ($\theta_1$)** | -0.38 (4.22) | - | -0.40 (6.47) | 0.89 (15.41) |
| **MA2 ($\theta_2$)** | - | - | -0.22 (3.82) | - |
| Seasonal Component [s = 7] ($P, D, Q$) | | | | |
| **SAR1 ($\phi_1^*$)** | 0.23 (3.02) | 0.27 (1.98) | 0.99 (204.37) | 0.27 (3.26) |
| **SAR2 ($\phi_2^*$)** | - | - | - | 0.23 (2.98) |
| **SMA1 ($\theta_1^*$)** | -0.87 (22.28) | - | -0.88 (23.67) | - |
| Goodness-of-fit | | | | |
| **LL** | -506.67 | -101.2 | -496.62 | -358.02 |
| **AIC** | 1025.13 | 210.4 | 1009.24 | 728.03 |
| **BIC** | 1047.3 | 218.28 | 1038.98 | 746.22 |
| Residuals–Fitting sample | | | | |
| **Ljung-Box (*P*-value)** | 0.51 | 0.79 | 0.56 | 0.45 |
| **MED** | 0.96 | 1.31 | 0.90 | 1.72 |
| **RMSE** | 1.30 | 1.62 | 1.21 | 2.48 |
| **Mean Error** | 0.04 | -0.04 | 0.02 | 0.2 |
| Residuals–Validation (out-of-sample data) | | | | |
| **MED** | 0.97 | 1.19 | 1.02 | 2.08 |
| **RMSE** | 1.30 | 1.46 | 1.37 | 2.55 |
| **Mean Error** | 0.50 | 0.37 | 0.82 | -0.63 |

properties, i.e. no evidence of autocorrelation was found, and the P-values of the Ljung-Box statistical test were all greater than 0.05. Results indicate an exceptional capability of the models to replicate recorded RRC using the HMI. RMSE values in the fitting stage fluctuated between 1.21 and 2.48 only, while MED ranged between 0.90 and 1.72 percentage points. Moreover, the quality of the predictions for the validation data was as high as the fitting stage, also obtaining remarkable goodness of fit. Once the models were validated, we employed them to predict RRC values on previously defined research periods.

## 4.1 London case study

The fitted and forecasted RRC for the London case study are shown in Fig 4A and 4B. Modelling results suggest a substantial under-reporting in ridership due to the free-bus policy enacted from 20 April to 24 May 2022 (see Fig 4B). The difference between recorded and

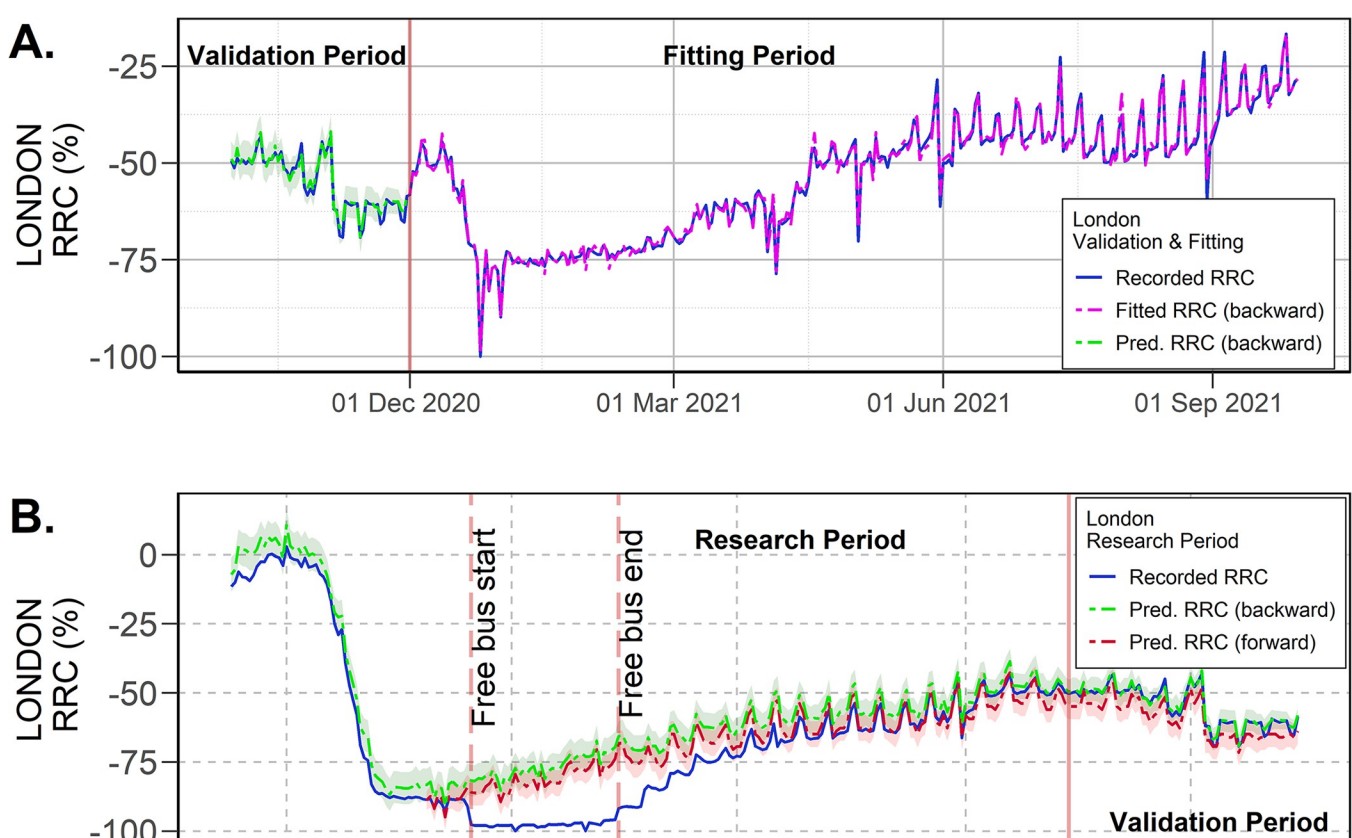

**Fig 4. Predicted RRC using calibrated HMI for London.** (A) Results for the fitting and validation period, (B) Forecasting for the research period (95% confidence intervals).

predicted RRC was at least 20 percentage points when the free-bus trip policy started on 20 April 2020. The predicted RRC (using forward and backward forecasting) coincided with the under-reporting magnitude, showing only that the forward forecasting generate a slightly more conservative prediction of the RRC. Predicted RRC suggested that the actual PT demand started to recover at the end of April 2020, not at the end of May, as the recorded RRC shows. Thereby, the dissimilarity between recorded and predicted RRC increased as PT demand started a slow and gradual recovery that ridership data did not take into account. The results also revealed that even when the free-bus policy finished on 24 May, the under-reporting in ridership continued for several months, gradually decreasing. In fact, according to Fig 4B, it took at least two months after the end of the free bus policy to observe the unification between the recorded and predicted RRC. This finding revealed a gradual adaptation process of PT users to return to normal payment behaviour after experiencing a free bus ride policy, which ridership data was unable to observe.

## 4.2 New York case study

Graphical results for the New York case study are provided in Fig 5A and 5B. Here, using HMI, it was possible to obtain an approximation of the actual ridership change in the city during the research period (see Fig 5B). To understand these results, two facts may be recalled from the New York data description: a) recorded RRC from 1 March to 30 September 2020 are

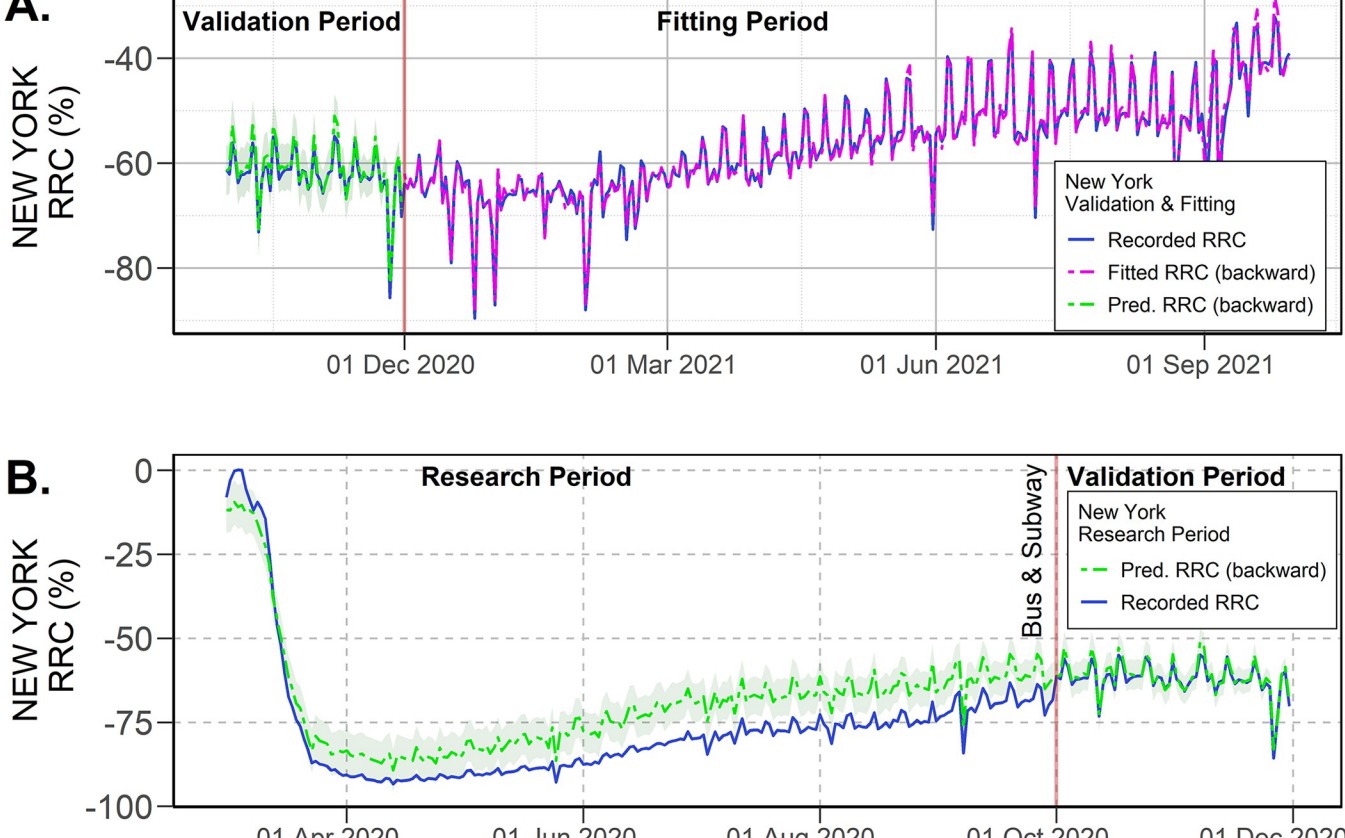

**Fig 5. Predicted RRC using calibrated HMI for New York.** (A) Results for the fitting and validation period, (B) Forecasting for the research period (95% confidence intervals).

based only on subway validations, and b) data from 1 October onward contain both bus and subway ridership data. Thereby, as the calibration between RRC-HMI is made when the complete data are available, the predicted RRC illustrates an approximation of the actual ridership in the New York MTA, as both bus and subway ridership would have been available. The results suggest that the actual changes in ridership in the system were lower in magnitude than the only-subway changes. Therefore, bus ridership should have experienced lower changes than the subway during the research period. In fact, the estimation showed that bus ridership was, on average, about 20 percentage points above the recorded relative subway changes for the period. It is interesting to notice that, as the predicted RRC is depicted below the recorded RRC on the first weeks of March 2020, the backward forecasting of RRC should be seen as a conservative approximation of the true RRC in terms of actual ridership recovery.

### 4.3 Santiago case study

Fitted and forecasted RRC for the Santiago case study are presented in Fig 6A, while Fig 6B zooms in on a fraction of the validation period. The results of the RRC prediction for the Chilean national referendum on 4 September 2022 showed that the HMI overestimated the recorded RRC (30.2% vs. 17.2%, see Fig 6B). This difference was much higher than any other prediction error observed in the validation period, suggesting that the reason for that difference was an exceptional overestimation in the mobility on PT hubs registered by the HMI.

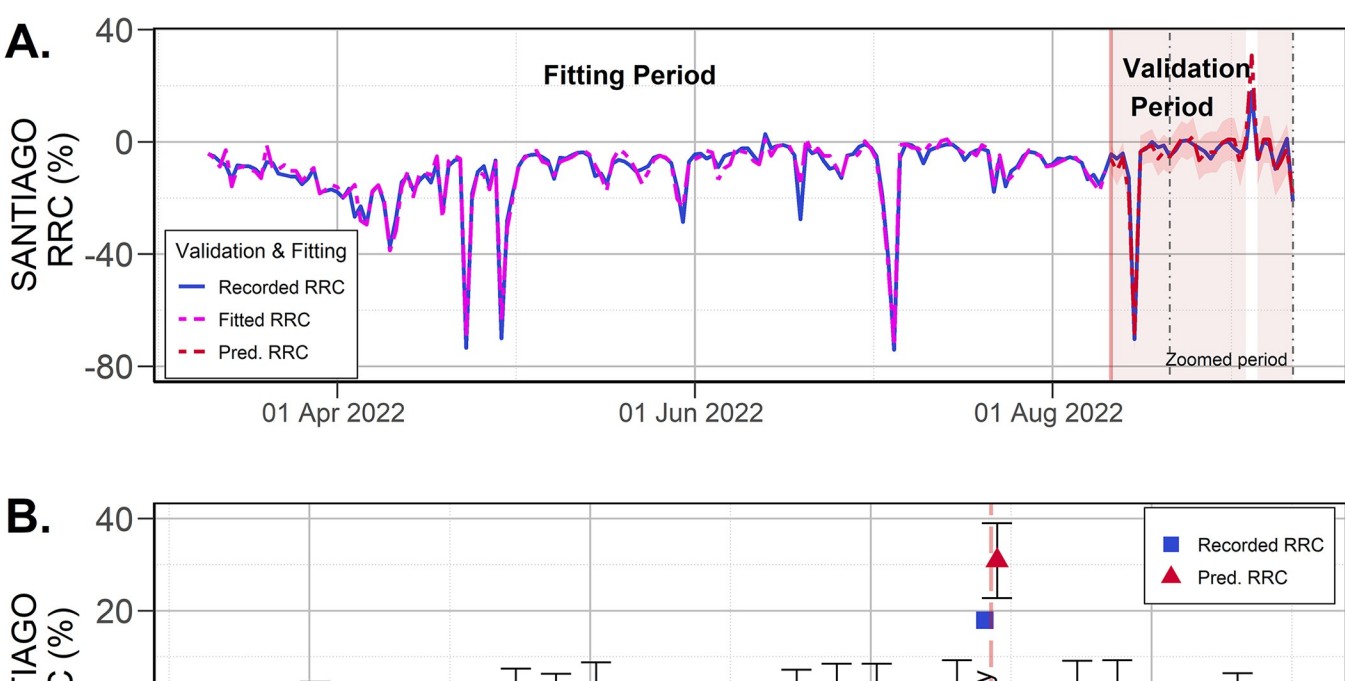

**Fig 6. Predicted RRC using calibrated HMI for Santiago.** (A) Results for the fitting and validation period, (B) Forecasting for the research period (95% confidence intervals).

Several elements that may have influenced the overestimation in the predicted RRC are hypothesised, related mainly to the nature of the HMI. For instance, a high HMI value may be associated with an increment in the time spent in PT stations due to higher waiting times of PT users caused by either a high demand or a limited PT supply. In fact, on 19 December 2021 (also an election day), Santiago's PT supply was severely criticised for the lack of bus services, low frequencies and unusually long waiting times. Interestingly, that day Santiago's HMI exhibited its highest value (57.1%) and highest difference with the RRC. An additional feasible cause of the HMI overestimation is linked to the exceptional nature of a national referendum, which involved millions of people travelling to their assigned locations. This generalised and exceptional number of people on the street may have affected the MI, for instance, by increasing pedestrian activity near PT hubs. In consequence, even when HMI successfully predicted RRC on regular days (including public holidays), it may be susceptible to registering higher mobility levels than the actual ridership on days with exceptional mobility.

## 5 Concluding discussion

Despite the extended use of aggregated mobility indices (AMIs) as proxy for the aggregate shifts in PT demand in the last years, existing research is inconclusive as to what extent they could replicate the changes recorded by actual PT ridership. The results reported here provide the first rigorous assessment of the capabilities of such indices to reproduce actual aggregate

shifts in PT demand. We conducted such assessment addressing the gaps of previous studies by: 1) establishing a common methodological approach for estimating relative mobility changes with different data sources, 2) considering a larger number of case studies and analysing differences in a more comprehensive study period and 3) exploring the complementary role of AMIs with ridership data. We summarise the result of their performance as follows:

- *Difference in relative changes between AMIs and ridership data (RRC)*: AMIs correctly captured the main changes in ridership levels, particularly for the first year of analysis (2020). When compared with ridership data, averages monthly differences of only 11.9 ± 3.6 and 11.6 ± 4.2 percentage points were found for the relative changes provided by Google (HMI) and Apple's Index (QMI) during 2020, respectively. While considering the daily analysis, average differences between 8.8 and 14.5 percentage points were observed. Even though these results suggest that AMIs tend to overestimate relative changes compared with ridership data, they greatly differ from previous studies, which reported differences between 30 to 50 percentage points for the same period [11,34,42]. The fact that previous studies have overlooked methodological differences between ridership data and AMIs in terms of their collection and definition would explain these discrepancies. For the following years, HMI kept a similar performance for all the study period, whilst QMI showed a substantial overestimation from April 2021.

- *Accuracy in replicating the direction of change of ridership*: The metric varied depending on the temporal granularity of the analysis. Based on the monthly similarity assessment, HMI and QMI correctly replicated as high as 85% of the direction changes. In the daily analysis, only the HMI kept a similar performance; QMI achieved slightly worse (61% to 73% overall). This difference in the QMI performance for the daily analysis has its root in the higher level of recovery in the number of PT queries on Fridays and Saturdays, which contrasts with the patterns recorded by ridership and HMI, which revealed higher recoveries on Saturdays and Sundays (See S2 Fig).

- *Overall consistency*: Strong evidence was found supporting a better performance of HMI relative to QMI. HMI showed a lower and more consistent mean distance with the changes reported by ridership data across the entire study period and a higher capability to replicate the direction of ridership change (between 10 and 20 percentage points more accurate). Additionally, the difference QMI-RRC presented five times the deviation observed between HMI-RRC. All these findings may imply that indices based on PT queries would be more prone to generate higher deviation and less accuracy in replicating ridership changes than GPS-based indices, particularly if extended periods are considered.

Different degrees of similarity were observed between the values of HMI and RRC among the case studies, with Sydney and London being the two cities where the highest similarities were found. We hypothesize that the definition of bus stop areas, fare evasion, coverage of PT services integrated into the AFC system and the variation in the PT infrastructure across the study period may have influenced how well HMI mimicked RRC. For instance, in the cases of London and Sydney, having well-established bus stop areas may have increased HMI's accuracy in accounting for changes in PT usage. Similarly, lower fare evasion and higher coverage of PT services in these two cities may have facilitated that ridership data reproduced the actual changes in the entire PT network. In contrast, for cities where only partial ridership was available (i.e. not all PT modes), the observed differences were higher (e.g. the Bogota case) or the differences observed followed different trends compared with other cases (e.g. Hong Kong). Since the characteristics and their effects discussed here remain speculative, further supplementary data collection efforts should be made to establish ground truth associations.

The overestimation in the QMI compared with the recovery in the actual ridership was observed for most of the cities analysed from April 2021, in moments when mobility restrictions were being eased. The reasons for this overestimation were likely the addition of the PT queries made by the new Apple Map app's users and the changes in the users' use behaviour of the Apple Map app. In particular, in this period users may have had a greater need for information on changes made to PT frequencies and services, which may have elevated the number of PT queries. These circumstances may have caused the increase observed in the QMI values immediately after the lifting of travel restrictions since this index was estimated considering a pre-COVID-19 baseline (which implicitly considers a pre-COVID-19 number of users and PT query behaviour). Hence, addressing the natural increase in the penetration of certain technology on which a QMI may be based, as well as the changes in the trend, may be relevant for future practical applications based on query data. This would improve the reliability of query indices for mid- and long-term analyses, especially when fixed baseline periods are considered.

Overall, two directions for potential uses of AMIs were identified: (a) providing a complementary characterisation of ridership changes and (b) providing supplementary information on the quality of PT services. Related to (a), in cities that do not have access to AFC systems, AMIs may play a key role in the analysis of PT systems, helping provide a refined characterisation of mobility trends to face global long-term events such as economic crises, pandemics/epidemics and conflicts, and local short-term events such as natural disasters, social unrest and transport supply disruptions. Such a characterisation is currently unavailable in these contexts, as existing traditional methods rely on information gathered by surveys, which provide restricted insights from small sample sizes and partial coverage of the consequences of the event (temporally and spatially). This contrasts with the capabilities of AMIs, which have the potential to provide continuous monitoring of the mobility over a region, registering the impacts of unanticipated events on PT demand and its resilience. In relation to cities that already have AFC systems, as we demonstrated in this study, AMIs may be useful when ridership data from ticketing is missing or of doubtful quality, such as in the cases of ticket-free riding days or when there are special periods where evasion is higher. In this regard, AMIs may offer a more far-reaching alternative to face this challenge than current methods based on manual passenger counting, motion and weight sensors, and CCTV cameras.

For the second category (b), the same raw ICT data used to estimate AMIs may be implemented to retrieve supplementary information on the quality of PT services. For example, GPS time on PT hubs could be analysed to study the time spent by owners of phone devices waiting for a PT service. This practical application has the potential to overcome the limitations of existing methods associated with travel surveys by providing a more dynamic, continuous and spatially richer characterisation of waiting times in PT systems. Regarding indices based on PT queries, there is potential in harnessing the dynamic fluctuations of information requested by travellers. For instance, an app query-based index may eventually be used to represent users' perceived level of reliability of the PT supply. This may help PT authorities take action regarding users' PT information needs. An atypical number of requests between specific O-D could be used to activate an immediate response from PT operators. The same data set could reveal whether there are PT service disruptions that could affect frequencies. PT query data may present advantages in identifying PT disruption compared with other novel approaches based on data from social media [47] since PT query data may be easily analysed based on variations in the query volumes. Interestingly, both data sources (PT query and social media platforms) can eventually be employed jointly to crosscheck information related to PT service disruption. In brief, the highest potential of AMIs is either their complementary role with existing smart card data or the provision of supplementary information on the quality of PT services.

The findings of this work indicate that AMIs based on data collected by smartphone apps have the potential to provide a reasonable proxy for the aggregate level shift in public transport (PT), particularly those that retrieve GPS traces, which also have the potential to provide supplementary information for PT. Nonetheless, many aspects of AMIs still need to be addressed in the future. The influence of the increasing number of users needs to be clarified, as well as the penetration rate needed to obtain reliable proxies. Additionally, the existing literature will greatly benefit from more transparency in how future AMIs are estimated. Ethical and privacy concerns are also elements that must be considered, as these data sets may reveal private user information and/or expose identifiable mobility traces. With a proper penetration rate, a ridership characterisation at a neighbourhood or more disaggregate level (e.g. at the level of PT hubs) may be available. These data would also allow observations to be made at a high granular temporal resolution, complementing the spatial heterogeneity in such data, eventually providing a rich representation of PT demand changes across the urban grid. To get to this stage, disaggregate data from ICT companies and App providers related to GPS traces and PT queries would need to be available (considering both temporal and spatial information). The decision of the private sector to make available these data may be motivated by the development of potential applications for the public transport sector. An assessment of the quality of these disaggregated data should be first conducted considering the desired spatial aggregation level (e.g. neighbourhood, census zone or PT hub). Such analyses should rely on a validation process that assesses the match between AMIs and ridership data at the chosen disaggregate level to investigate AMIs' data appropriateness. Special attention should be paid to identifying characteristics of the disaggregate zones that may explain the capability of the AMI to mimic ridership across the city (e.g. availability of PT infrastructure, PT demand characteristics). This analysis would allow the possibility to improve reports on particular zones, increasing the reliability of AMIs to represent the changes of PT demand across the city.

Considering the current and future urban challenges, the importance of mobility data availability transcends the COVID-19 pandemic. With that in mind, the main contribution of this work is having proved that AMIs based on a regular smartphone use may be used to generate a reasonable approximation of the actual aggregate PT demand changes. The results of this paper support the need for replacing discontinued AMIs provided during the COVID-19 pandemic by proposing new AMIs based on similar data sets. For instance, it has been recently demonstrated that it is possible to replicate big tech companies' AMIs using GPS traces collected by an emergency smartphone alert app [19]. Regarding the level of PT queries, these data are currently collected by many private companies and PT operators that run PT trip planners locally or globally. Additionally, future research should focus not only on validating new proxies for PT demand based on data collected by mobile phone apps, but also on comprehensively integrating these emerging datasets with traditional ones.

## Supporting information

**S1 Fig. Average mobility per day of the week by data set, baseline period.**
(TIF)

**S2 Fig. Weekly periodicity pattern in by index for the London case study.**
(TIF)

**S1 Table. Monthly and daily datasets.**
(XLSX)

## Author Contributions

**Conceptualization:** Maximiliano Lizana, Charisma Choudhury, David Watling.

**Data curation:** Maximiliano Lizana.

**Formal analysis:** Maximiliano Lizana.

**Investigation:** Maximiliano Lizana.

**Methodology:** Maximiliano Lizana, Charisma Choudhury, David Watling.

**Software:** Maximiliano Lizana.

**Supervision:** Charisma Choudhury, David Watling.

**Validation:** Maximiliano Lizana.

**Visualization:** Maximiliano Lizana.

**Writing – original draft:** Maximiliano Lizana.

**Writing – review & editing:** Maximiliano Lizana, Charisma Choudhury, David Watling.

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
