## [Decision Letter · Decision Letter 0]

27 Oct 2023

PONE-D-23-29764Investigating the potential of aggregated mobility indices for inferring public transport ridership changesPLOS ONE

Dear Dr. Choudhury,

Thank you for submitting your manuscript to PLOS ONE. After careful consideration, we feel that it has merit but does not fully meet PLOS ONE’s publication criteria as it currently stands. Therefore, we invite you to submit a revised version of the manuscript that addresses the points raised during the review process.

Two experts in the field of travel behavior have evaluated your manuscript and provided their comments. Both reviewers would like to see some further improvements in your article. In particular, potential practical applications can be discussed. In addition the results/ findings could also be further elaborated.   

We look forward to receiving your revised manuscript.

Kind regards,

Charitha Dias

Academic Editor

PLOS ONE

Journal Requirements:

2. In your Methods section, please include additional information about your dataset and ensure that you have included a statement specifying whether the collection and analysis method complied with the terms and conditions for the source of the data.

The funding for this research has been provided by the Chilean Agency of Research and Development (ANID) through the Becas Chile scholarship. Professor Charisma Choudhury’s time was supported by the UKRI Future Leader

Fellowship [MR/T020423/1].

Reviewers' comments:

Reviewer's Responses to Questions

**Comments to the Author**

1. Is the manuscript technically sound, and do the data support the conclusions?

Reviewer #1: Yes

Reviewer #2: Yes

2. Has the statistical analysis been performed appropriately and rigorously? 

Reviewer #1: Yes

Reviewer #2: Yes

3. Have the authors made all data underlying the findings in their manuscript fully available?

Reviewer #1: Yes

Reviewer #2: Yes

4. Is the manuscript presented in an intelligible fashion and written in standard English?

Reviewer #1: Yes

Reviewer #2: Yes

5. Review Comments to the Author

Reviewer #1: The research underlying this manuscript investigates an important and relevant topic in public transport planning and policy: the potential of aggregated mobility indices - such as Google's Community Mobility Reports - to infer public transport ridership changes. The manuscript is well written and structured. A highly rigorous methodology is employed and the results are presented clearly. An impressive feature of the research is that 12 cities worldwide were covered. Covering even one city at such a detailed level would have still provided useful insights. An additional strength is that three different measures (MED, COS, TSI) are used to assess similarity, as opposed to just one measures. Given the high quality of the manuscript, I only have relatively minor comments, as follows:

1. In Section 3.1, the results of the monthly analysis are presented. Here, it is found that London and Sydney had the greatest match between HMI and RRC. This begs the question - why is this the case? There may not be a clear answer for this, and if so, the authors should still acknowledge possible reasons for the small difference relative to other cities.

2. Also in Section 3.1, it is found that QMI showed a general increase from HMI between April-August 2021 - again, why is this the case? It would be helpful if the authors could comment on this, either at this point in the paper or later in the discussion/conclusion.

3. In Section 5, Concluding discussion, there is some discussion about AMIs playing a key role in the analysis of PT systems in cities that do not have access to AFC systems. It is noted that with a proper penetration rate, ridership at a more disaggregate level may be available. It would be helpful at this point if the authors could discuss how we could/should get to this stage and what would be required to achieve this.

4. Slightly later in Section 5, the authors discuss how AMIs could be useful when ridership data from ticketing is missing or of doubtful quality - this is an excellent point, but could or should AMIs eventually replace traditional ridership data sources such as ticketing? If so, what should be the path to achieving this?

Minor comments:

- Abstract: where "12 cities" are mentioned, it would be good to at least say "12 cities worldwide" so that readers don't necessarily assume they are 12 cities from within the same country.

- Section 1.3, page 7: where "12 cities worldwide" are mentioned, it would be helpful here to add "from 8 countries" to provide some additional context. Otherwise, readers have to wait until Section 2 (page 9) to find out how many countries were covered.

- Table 1, page 9: "Sidney" should be "Sydney". Also, please check the PT authority name for Sydney which I think should be Transport for NSW (Opal is the name of the ticketing system).

- Section 3.1, page 16: "Sidney" should be "Sydney".

- Table 3, page 17: AMIs, RRC, MED, STI and COS should all be at least spelt out in full beneath the table.

- Page 29, line 605: "used" rather than "use"

- Figures 2-6 are all very fuzzy - it would be better if higher resolution images could be used here.

Reviewer #2: This paper is interesting, technically sound, and quite well written. I have only a few minor suggestions for further improvements regarding the practical applications. Currently there are no baseline methods for comparison. For example, what could be the traditional methods to address the problems presented in the practical applications? This will help demonstrate the effectiveness of the proposed methods. It would be expected that the proposed methods would perform well with disruptive events (applications 1 and 3) whereas traditional methods, even one based on historical trend, would perform well in application 2. At least some discussions around these issues will improve the value of the paper.

6. PLOS authors have the option to publish the peer review history of their article (what does this mean?). If published, this will include your full peer review and any attached files.

Reviewer #1: No

Reviewer #2: No

---

## [Author Response · Author response to Decision Letter 0]

5 Dec 2023

Reviewer #1: 

The research underlying this manuscript investigates an important and relevant topic in public transport planning and policy: the potential of aggregated mobility indices - such as Google's Community Mobility Reports - to infer public transport ridership changes. The manuscript is well written and structured. A highly rigorous methodology is employed and the results are presented clearly. An impressive feature of the research is that 12 cities worldwide were covered. Covering even one city at such a detailed level would have still provided useful insights. An additional strength is that three different measures (MED, COS, TSI) are used to assess similarity, as opposed to just one measures. Given the high quality of the manuscript, I only have relatively minor comments, as follows:

Response:

Thanks for the positive feedback and constructive suggestions. We have carefully addressed each comment received and made the necessary revisions to the manuscript. We believe these revisions have strengthened the manuscript by clarifying the practical implications of its findings. A point-by-point response to each of the comments received can be found below.

1. In Section 3.1, the results of the monthly analysis are presented. Here, it is found that London and Sydney had the greatest match between HMI and RRC. This begs the question - why is this the case? There may not be a clear answer for this, and if so, the authors should still acknowledge possible reasons for the small difference relative to other cities.

Response:

We appreciate the observation and recognise the importance of extending the discussion about this finding. We have discussed now the possible reasons for this result in the discussion section (lines 573-586):

“Different degrees of similarity were observed between the values of HMI and RRC among the case studies, with Sydney and London being the two cities where the highest similarities were found. We hypothesize that the definition of bus stop areas, fare evasion, coverage of PT services integrated into the AFC system and the variation in the PT infrastructure across the study period may have influenced how well HMI mimicked RRC. For instance, in the cases of London and Sydney, having well-established bus stop areas may have increased HMI’s accuracy in accounting for changes in PT usage. Similarly, lower fare evasion and higher coverage of PT services in these two cities may have facilitated that ridership data reproduced the actual changes in the entire PT network. In contrast, for cities where only partial ridership was available (i.e. not all PT modes), the observed differences were higher (e.g. the Bogota case) or the differences observed followed different trends compared with other cases (e.g. Hong Kong). Since the characteristics and their effects discussed here remain speculative, further supplementary data collection efforts should be made to establish ground truth associations.”

2. Also in Section 3.1, it is found that QMI showed a general increase from HMI between April-August 2021 - again, why is this the case? It would be helpful if the authors could comment on this, either at this point in the paper or later in the discussion/conclusion.

Response:

Thanks for pointing out the question. We now discussed potential explanations in Section 5 (lines 587-600): 

“The overestimation in the QMI compared with the recovery in the actual ridership was observed for most of the cities analysed from April 2021, in moments when mobility restrictions were being eased. The reasons for this overestimation were likely the addition of the PT queries made by the new Apple Map app’s users and the changes in the users’ use behaviour of the Apple Map app. In particular, in this period users may have had a greater need for information on changes made to PT frequencies and services, which may have elevated the number of PT queries. These circumstances may have caused the increase observed in the QMI values immediately after the lifting of travel restrictions since this index was estimated considering a pre-COVID-19 baseline (which implicitly considers a pre-COVID-19 number of users and PT query behaviour). Hence, addressing the natural increase in the penetration of certain technology on which a QMI may be based, as well as the changes in the trend, may be relevant for future practical applications based on query data. This would improve the reliability of query indices for mid- and long-term analyses, especially when fixed baseline periods are considered.” 

3. In Section 5, Concluding discussion, there is some discussion about AMIs playing a key role in the analysis of PT systems in cities that do not have access to AFC systems. It is noted that with a proper penetration rate, ridership at a more disaggregate level may be available. It would be helpful at this point if the authors could discuss how we could/should get to this stage and what would be required to achieve this.

Response:

We appreciate the opportunity to highlight the next steps in a potential disaggregate application. A brief discussion about this is now provided between lines 652 and 665. Note that we slightly changed the location of this topic to maintain consistency in each paragraph.

“To get to this stage, disaggregate data from ICT companies and App providers related to GPS traces and PT queries would need to be available (considering both temporal and spatial information). The decision of the private sector to make available these data may be motivated by the development of potential applications for the public transport sector. An assessment of the quality of these disaggregated data should be first conducted considering the desired spatial aggregation level (e.g. neighbourhood, census zone or PT hub). Such analyses should rely on a validation process that assesses the match between AMIs and ridership data at the chosen disaggregate level to investigate AMIs’ data appropriateness. Special attention should be paid to identifying characteristics of the disaggregate zones that may explain the capability of the AMI to mimic ridership across the city (e.g. availability of PT infrastructure, PT demand characteristics). This analysis would allow the possibility to improve reports on particular zones, increasing the reliability of AMIs to represent the changes of PT demand across the city.”

4. Slightly later in Section 5, the authors discuss how AMIs could be useful when ridership data from ticketing is missing or of doubtful quality - this is an excellent point, but could or should AMIs eventually replace traditional ridership data sources such as ticketing? If so, what should be the path to achieving this?

Response:

Thanks for the question. We do not believe it is possible to conclude, based on our results, that AMIs may eventually replace ticketing. Although our work presents promising practical applications of AMIs in the PT domain, the applied approach and results only allow the recognition of a complementary role of AMIs with traditional ridership data, regarding identifying missing ticketing. In particular, the approach employed in this study relied on the combination of AMIs and ridership data from stable periods to reveal missing ticketing. This combination is needed as AMIs are estimated as a relative measure that accounts only for relative changes in PT demand, and they do not provide information for an absolute metric (like the total number of trips).

Minor comments:

- Abstract: where "12 cities" are mentioned, it would be good to at least say "12 cities worldwide" so that readers don't necessarily assume they are 12 cities from within the same country.

Response: This is a very useful suggestion. The modification was made following the reviewer’s advice.

- Section 1.3, page 7: where "12 cities worldwide" are mentioned, it would be helpful here to add "from 8 countries" to provide some additional context. Otherwise, readers have to wait until Section 2 (page 9) to find out how many countries were covered.

Response: The clarification is now added in the revised version of the manuscript.

- Table 1, page 9: "Sidney" should be "Sydney". Also, please check the PT authority name for Sydney which I think should be Transport for NSW (Opal is the name of the ticketing system). Section 3.1, page 16: "Sidney" should be "Sydney".

Response: We apologise for the typo. We have checked the manuscript to remove any similar mistake. The Reviewer is also right about the name of the PT authority. Therefore, we have corrected it accordingly in the manuscript. Thank you very much.

- Table 3, page 17: AMIs, RRC, MED, STI and COS should all be at least spelt out in full beneath the table.

Response: All the mentioned indices are now spelt out in full beneath Table 3.

- Page 29, line 605: "used" rather than "use".

Response: Many thanks for the observation. We have corrected it in the revised version.

- Fig. 2-6 are all very fuzzy-it would be better if higher resolution images could be used here.

Response: 

We have improved the image resolution of all figures. We have also checked them using the digital diagnostic tool suggested by the Journal. If low-resolution figures still appear in the revised version of the manuscript, it is likely that by default the file shared in the review process considers a low-resolution standard. We will carefully follow this issue during the production stage to ensure the quality of the figures provided. Thank you for letting us know it.

Reviewer #2: 

This paper is interesting, technically sound, and quite well written. I have only a few minor suggestions for further improvements regarding the practical applications.

Response:

We thank the Reviewer #2 for the positive feedback given to our work. We also appreciate Reviewer #2’ comments, which helped us to highlight the relevance of our findings regarding their practical applications.

Currently there are no baseline methods for comparison. For example, what could be the traditional methods to address the problems presented in the practical applications? This will help demonstrate the effectiveness of the proposed methods. It would be expected that the proposed methods would perform well with disruptive events (applications 1 and 3) whereas traditional methods, even one based on historical trend, would perform well in application 2. At least some discussions around these issues will improve the value of the paper.

Response:

We appreciate the comment of the Reviewer, which helped us to highlight the potential of aggregate mobility indices (AMIs) relative to traditional/current methods. We have included the role of alternative methods in each of the problems presented after the corresponding practical application of the AMIs (lines 601-638).

“Related to the first one (a), in cities that do not have access to AFC systems, AMIs may play a key role in the analysis of PT systems, helping provide a refined characterisation of mobility trends to face global long-term … and local short-term events …. Such a characterisation is currently unavailable in these contexts, as existing traditional methods rely on information gathered by surveys, which provide restricted insights from small sample sizes and partial coverage of the consequences of the event (temporally and spatially). This contrasts with the capabilities of AMIs, which have the potential to provide continuous monitoring of the mobility over a region, registering the impacts of unanticipated events on PT demand and its resilience. In relation to cities that already have AFC systems, as we demonstrated in this study, AMIs may be useful when ridership data from ticketing is missing or of doubtful quality, such as in the cases of ticket-free riding days or when there are special periods where evasion is higher. In this regard, AMIs may offer a more far-reaching alternative to face this challenge than current methods based on manual passenger counting, motion and weight sensors, and CCTV cameras.

For the second category (b), the same raw ICT data used to estimate AMIs may be implemented to retrieve supplementary information on the quality of PT services. For example, GPS time on PT hubs could be analysed to study the time spent by owners of phone devices waiting for a PT service. This practical application has the potential to overcome the limitations of existing methods associated with travel surveys by providing a more dynamic, continuous and spatially richer characterisation of waiting times in PT systems. Regarding indices based on PT queries, there is potential in harnessing the dynamic fluctuations of information requested by travellers. For instance, an app query-based index may eventually be used to represent users’ perceived level of reliability of the PT supply. This may help PT authorities take action regarding users’ PT information needs. An atypical number of requests between specific O-D could be used to activate an immediate response from PT operators. The same data set could reveal whether there are PT service disruptions that could affect frequencies. PT query data may present advantages in identifying PT disruption compared with other innovative approaches based on data from social media [47] since PT query data may be easily analysed based on variations in the query volumes. Interestingly, both data sources (PT query and social media platforms) can eventually be employed jointly to crosscheck information related to PT service disruption. In brief, the highest potential of AMIs is either their complementary role with existing smart card data or the provision of supplementary information on the quality of PT services.

47. Chan R, Schofer J. Role of Social Media in Communicating Transit Disruptions. Transp. Res. Rec. 2014; 2415: 145-151

 

Journal Requirements:

Response:

We have fully checked the style templates and made the corresponding amends in the revised manuscript. Among other changes, we have:

• Added the corresponding author’s initials in parentheses after the email.

• Checked the format of how multiple figures and tables were cited.

• Checked the sentence case format in all headings.

• Located all figure and table captions directly after the paragraph in which they were cited.

2. In your Methods section, please include additional information about your dataset and ensure that you have included a statement specifying whether the collection and analysis method complied with the terms and conditions for the source of the data.

Response:

A detailed description of the datasets used in this study is provided in the Method section. This included the sub-sections “Data” and “Establishing a common definition” (and the associated Fig 1, Table 1 and 2). An early characterization of the aggregate mobility indices is also presented in the introduction (Section 3.1), as it was required to define the gaps of this study. We have pointed out this in the method section (line 176-177).

We have also added the statement requested at the end of the first paragraph of Section 2.1, lines 194-197. “This work employed publicly available data, whose use complied with the terms and conditions for each source. Further details of the terms and conditions can be found directly in the web pages of each source using the links provided in the Supporting files (S1 Table).”

3. Thank you for stating in your Funding Statement: The funding for this research has been provided by the Chilean Agency of Research and Development (ANID) through the Becas Chile scholarship. Professor Charisma Choudhury’s time was supported by the UKRI Future Leader Fellowship [MR/T020423/1].

Response:

We can confirm that all the funding or sources of support (internal and external) associated with this study are acknowledged in the Funding Statement. The Funding Statement was modified according to the guide for authors and added to the cover letter.

Response:

We have updated the Data Availability statement in the cover letter to highlight that the mentioned minimal data set is available in the supporting information files. 

Response:

The reference list format was checked according to the suggested style of PLOS ONE. As far as the authors know, no retracted paper is in the reference list. Only one additional reference was added to the reference list during the revision of the manuscript [47].

---

## [Decision Letter · Decision Letter 1]

18 Dec 2023

Investigating the potential of aggregated mobility indices for inferring public transport ridership changes

PONE-D-23-29764R1

Dear Dr. Choudhury,

We’re pleased to inform you that your manuscript has been judged scientifically suitable for publication and will be formally accepted for publication once it meets all outstanding technical requirements.

Kind regards,

Charitha Dias

Academic Editor

PLOS ONE

Additional Editor Comments (optional):

Reviewers' comments:

Reviewer's Responses to Questions

**Comments to the Author**

1. If the authors have adequately addressed your comments raised in a previous round of review and you feel that this manuscript is now acceptable for publication, you may indicate that here to bypass the “Comments to the Author” section, enter your conflict of interest statement in the “Confidential to Editor” section, and submit your "Accept" recommendation.

Reviewer #1: All comments have been addressed

Reviewer #2: All comments have been addressed

2. Is the manuscript technically sound, and do the data support the conclusions?

Reviewer #1: Yes

Reviewer #2: Yes

3. Has the statistical analysis been performed appropriately and rigorously? 

Reviewer #1: Yes

Reviewer #2: Yes

4. Have the authors made all data underlying the findings in their manuscript fully available?

Reviewer #1: Yes

Reviewer #2: Yes

5. Is the manuscript presented in an intelligible fashion and written in standard English?

Reviewer #1: Yes

Reviewer #2: Yes

6. Review Comments to the Author

Reviewer #1: Thank you very much for addressing the comments from the reviewers in such a careful and considered manner. I have no further comments on the manuscript and would like to wish the authors all the very best with disseminating their research findings.

Reviewer #2: The authors have addressed all comments and improved the paper significantly. The paper is now publishable

7. PLOS authors have the option to publish the peer review history of their article (what does this mean?). If published, this will include your full peer review and any attached files.

Reviewer #1: No

Reviewer #2: No

---

## [Editor Report · Acceptance letter]

28 Dec 2023

PONE-D-23-29764R1 

PLOS ONE

Dear Dr. Choudhury, 

I'm pleased to inform you that your manuscript has been deemed suitable for publication in PLOS ONE. Congratulations! Your manuscript is now being handed over to our production team.

Kind regards, 

on behalf of

Dr. Charitha Dias 

Academic Editor

PLOS ONE